# Calcium Sulfite Solids Activated by Iron for Enhancing As(III) Oxidation in Water

**DOI:** 10.3390/molecules26041154

**Published:** 2021-02-21

**Authors:** Minjuan Cai, Sen Quan, Jinjun Li, Feng Wu, Gilles Mailhot

**Affiliations:** 1Hubei Key Lab of Biomass Resource Chemistry and Environmental Biotechnology, School of Resources and Environmental Science, Wuhan University, Wuhan 430079, China; caiminjuan@whu.edu.cn (M.C.); lijinjun@whu.edu.cn (J.L.); fengwu@whu.edu.cn (F.W.); 2Hubei Academy of Environmental Science, Wuhan 430072, China; 3CNRS, SIGMA Clermont, Institut de Chimie de Clermont-Ferrand, Université Clermont Auvergne, F-63000 Clermont–Ferrand, France; gilles.mailhot@uca.fr

**Keywords:** desulfurized gypsum, Fe(III), CaSO_3_, As(III), resource utilization

## Abstract

Desulfurized gypsum (DG) as a soil modifier imparts it with bulk solid sulfite. The Fe(III)–sulfite process in the liquid phase has shown great potential for the rapid removal of As(III), but the performance and mechanism of this process using DG as a sulfite source in aqueous solution remains unclear. In this work, employing solid CaSO_3_ as a source of SO_3_^2−^, we have studied the effects of different conditions (e.g., pH, Fe dosage, sulfite dosage) on As(III) oxidation in the Fe(III)–CaSO_3_ system. The results show that 72.1% of As(III) was removed from solution by centrifugal treatment for 60 min at near-neutral pH. Quenching experiments have indicated that oxidation efficiencies of As(III) are due at 67.5% to HO^•^, 17.5% to SO_5_^•−^ and 15% to SO_4_^•−^. This finding may have promising implications in developing a new cost-effective technology for the treatment of arsenic-containing water using DG.

## 1. Introduction

The wet desulfurization process has significant advantages of fast reaction and high efficiency, and has been widely used in China. However, desulfurized gypsum (DG) cannot be fully utilized, and its disposal constitutes a waste of resources [1,2]. Currently, DG is mainly used as a soil conditioner [3], serving to improve certain properties of soil, such as its pH, water absorption, water retention, and so on. The potential oxidative ability of DG towards heavy metals or organic pollutants in soil-containing transition metals has rarely been reported [4,5,6,7].

Recently, several investigations have shown that sulfate radical (SO_4_^•−^)-based advanced oxidation processes work well for the removal of organic and inorganic pollutants in groundwater and wastewater treatment [8,9,10]. The commonly used oxidants include persulfate (PS), peroxymonosulfate (PMS), and sulfite. Compared to PS and PMS, sulfite has significant advantages of low toxicity, low cost, and easy preparation, thus making it both environmentally friendly and economic [11]. Transition metal systems, such as Fe(II/III), have been reported to activate sulfite, such that it has a good oxidation effect on arsenic and organic pollutants in water [12,13,14,15,16]. Additionally, iron-based nanomaterials of high superficial activity also had a good effect on removing heavy metal pollutants in the environment [17,18]. In the Fe(II/III)-sulfite system, S(IV) can be catalytically oxidized under certain conditions to produce a series of oxysulfur species, including sulfite radical (SO_3_^•−^) and sulfate radical (SO_4_^•−^) [19,20,21]. An intrinsic mechanism has been proposed for this system, which includes the following reactions (Equations (1)–(8)) [22,23,24,25,26]:Fe^2^^+^ + HSO_3_^−^ ⇌ FeHSO_3_^+^ (rapid equilibration)(1)
4FeHSO_3_^+^ + O_2_ → 4FeSO_3_^+^ + 2H_2_O(2)
FeSO_3_^+^ → Fe^2+^ + SO_3_^•−^ (k_f_ = 0.19 s^−1^)(3)
SO_3_^•−^ + O_2_ → SO_5_^•^^−^ (k_4_ < 10^9^ mol^−1^ L s^−1^)(4)
SO_5_^•−^ + HSO_3_^−^ → SO_3_^•−^ + HSO_5_^−^ (k_5_ = (10^4^–10^7^) mol^−1^ L s^−1^)(5)
Fe^2+^ + HSO_5_^−^ → SO_4_^•−^ + Fe^3+^ + OH^−^ (k_6_ = (10^4^–10^7^) mol^−1^ L s^−1^)(6)
Fe^3+^ + HSO_3_^−^ ⇌ FeSO_3_^+^ + H^+^ (logk_7_ = 2.45)(7)
SO_5_^•−^ + HSO_3_^−^ → SO_4_^2−^ + SO_4_^•−^ + H^+^ (k_8_ = (10^4^–10^7^) mol^−1^ L s^−1^)(8)

In actual application processes, soluble SO_3_^2−^ is consumed very rapidly. Due to the slow dissolution of solid CaSO_3_, the concentration of soluble SO_3_^2−^ in the solution can keep in an appropriate range [27]. Although Fe(II) activated homogeneous sulfite has been fully studied and reported, the activation mechanism of heterogeneous calcium sulfite remains unclear. In summary, it was necessary to research and develop new utilization methods of calcium sulfite waste.

In this work, CaSO_3_ has been directly employed as a sulfite donor to assess the oxidation capability of the Fe(III)–sulfite system. The effects of CaSO_3_/Fe(III) concentration, pH, and initial concentrations of As(III) have been investigated. Free radical quenching experiments were then performed to determine the contributions of reactive species in this system. Our work may contribute to an understanding of the main mechanisms of As(III) removal using the iron–CaSO_3_ system and provide a theoretical basis for the use of desulfurized gypsum as a soil conditioner.

## 2. Results and Discussion

### 2.1. Control Experiments

As shown in Figure 1, As(tot) remained unchanged in both the control experiments and the Fe(III)-CaSO_3_ system experimental group, so the degradation of As(III) was due to the conversion to As(V) by oxidation reaction. The deployment of Fe(III) or CaSO_3_ alone led to 20% degradation of As(III) in 1 h, which is in obvious contrast to the combination of Fe(III) and CaSO_3_ (72%).

Fe(III) in the solution mainly existed in the form of colloid at near-neutral pH and thus adsorbed As(III) through surface complexation. As(III) can then be oxidized by electron transfer from As(III) to Fe(III) induced by radiation absorption via ligand-to-metal charge transfer [28]. The decrease in As(III) concentration in the presence of Fe(III) in the absence of sulfite may be due to radical oxidation pathway of As(III) and adsorption on colloidal ferric hydroxides followed by oxidation [19]. The decrease in As(III) concentration in the presence of sulfite alone may be due to the auto-oxidation reaction of SO_3_^2−^ on the surface of CaSO_3_ solid surface (>Ca(II)-SO_3_^2−^ in Equations (9)–(13)) and the co-oxidation of SO_3_^2−^ with As (III) under alkaline conditions on the surface of the CaSO_3_ solid (Equations (14) and (15)).
>CaSO_3_ → >Ca(II)-SO_3_^2−^(9)
>Ca(II)-SO_3_^2−^ + O_2_ → >Ca(II)-SO_3_^2−^-O_2_(10)
>Ca(II)-SO_3_^2−^-O_2_ → >Ca(II)-SO_3_^•–^ + O_2_^•−^(11)
>Ca(II)-SO_3_^•−^ + O_2_ → >Ca(II)-SO_5_^•−^(12)
>Ca(II)-SO_5_^•−^ + As(III) → >Ca(II)-As(V) + SO_4_^2−^(13)
>CaSO_3_ + H_2_O → >Ca(II)-OH^−^ + SO_3_^2−^ + H^+^(14)
>Ca(II)-OH^−^ (mediated reaction): SO_3_^2−^ + As(III) + O_2_ → SO_4_^2−^ + As(V)(15)

When Fe(III) and CaSO_3_ solid powders were added simultaneously, the reaction rate of the system increased rapidly. This is basically consistent with the previous research results on the Fe(III)-sulfite system [14,20,21], suggesting that the activation of SO_3_^2−^ may occur either in the liquid phase or on the surface of CaSO_3_ particles (Equations (16) and (17)). Considering that irons can easily form hydroxide precipitates at near neutral pH and aggregate with CaSO_3_ to form composite particles, the activation of SO_3_^2−^ on the solid surface may also be important.
>Ca(II)-SO_3_^2−^ + Fe(III) → >Ca(II)-SO_3_^2−^-Fe(III)(16)
>Ca(II)-SO_3_^2−^-Fe(III) → >Ca(II)-SO_3_^•−^ + Fe(II)(17)

### 2.2. Effects of Fe(III) and CaSO_3_ Dosages on As(III) Oxidation

The effects of Fe(III) and CaSO_3_ dosages on As(III) oxidation were investigated (Figure 2). For optimizing Fe(III) dosage, 0.5 mM Fe(III) achieved the best As(III) oxidation. When Fe(III) concentrations exceed this value, As(III) oxidation was conversely inhibited. This can be explained by the fact that a greater amount of Fe(OH)_3_ colloid might be produced in the reaction system, which would inhibit the dissolution and release of CaSO_3_ and the activation ability of Fe(III) towards HSO_3_^−^. Besides, a large amount of Fe(II) generated by the initial reactions (Equations (1)–(3)) could also consume SO_5_^•−^/SO_4_^•−^ in the solution (Equations (18) and (19)). However, due to the presence of excess CaSO_3_, HSO_3_^−^ continues to slowly dissolve, and the oxidation efficiency of As(III) does not decrease steadily with the increase in Fe(III) concentration. Hence, Fe(III) concentration should not be the only limiting factor for As(III) oxidation in the Fe(III)-CaSO_3_ system.
SO_5_^•−^ + Fe^2+^ + H^+^ → HSO_5_^−^ + Fe^3+^(18)
SO_4_^•−^ + Fe^2+^ → SO_4_^2−^ + Fe^3+^(19)

Meanwhile, for optimizing CaSO_3_ dosages, 30 mM CaSO_3_ achieved the best As(III) oxidation. When CaSO_3_ concentration exceed this value, the concentration of S(IV) is much higher than that of As(III) in the solution and As(III) oxidation was inhibited. The reasons for the inhibition of As(III) oxidation may be that S(IV) on the surface of CaSO_3_ may compete with As(III) for SO_4_^•−^ [Equation (20)]. Therefore, the removal of SO_4_^•−^ by excess S(IV) may be a reason for the inhibition of As(III) oxidation. Zhou et al. [18] delineated the reactions of SO_5_^•−^/SO_4_^•−^ in the Fe(III)–S(IV) system, indicating that an excess of S(IV) or Fe(II) may inhibit the reactions of SO_5_^•−^/SO_4_^•−^. At the same time, excess CaSO_3_ may lead to a large number of particles around Fe(III), which would inactivate it and reduce its ability to activate S(IV).
SO_4_^•−^ + HSO_3_^−^ → HSO_4_^−^ + SO_3_^•−^(20)

### 2.3. Effect of pH on As(III) Oxidation

To investigate the effect of pH on the Fe(III)-CaSO_3_ system, experiments were conducted at pH 4, 6, 8, and 10 (Figure 3). It was found that As(III) oxidation efficiency varied from 63% at pH 10 to 80% at pH 4 within 60 min and when at pH 6 and 7, the efficiency of the system had no obvious difference, reaching about 75%.

Lower pH values led to more efficient As(III) oxidation, consistent with conclusions drawn for the Fe(III)-Na_2_SO_3_ system [12]. This may be because, at near-neutral pH, the Fe in the solution mainly exists in the form of Fe(OH)_2_^+^ or Fe(OH)_3_, and forms a complex with sulfite [19]. Fe(III) and Fe(II) in the reaction solution can be rapidly converted, according to Equations (3) and (7), which is also a key step affecting the reaction rate in the Fe(III)-Na_2_SO_3_ system. When the pH approaches alkaline, the system still maintained good oxidation efficiency, probably because the interconversion of SO_4_^•−^ to HO^•^ (Equation (21)) is favored at pH > 8.5 and the direct activation of sulfite by alkalion [29].
SO_4_^•−^ + OH^−^ → SO_4_^2−^ + HO^•^ (alkaline pH)(21)

### 2.4. Effect of the Initial Concentration of As(III) on Its Oxidation

As(III) oxidation processes followed Langmuir–Hinshelwood (L–H) kinetics (Equation (22) [30].
(22)$$r_{0}\;=\;\frac{K_{L-H}\;\times \;K\;\times \;C_{0}}{1\;+\;K_{L-H}\;\times \;C_{0}}$$

The kinetics of As(III) oxidation in the Fe(III)-CaSO_3_ system was studied by the initial rate method. The initial rate r_0_ was taken as the average value of the change of As(III) concentration over the initial period. The initial oxidation rate of As(III) increased with its initial concentration (Figure 4). When the initial As(III) concentration is higher, the As(III) oxidation rate continues to increase. The relationship between r_0_ and C_0_ for As(III) oxidation followed the Langmuir–Hinshelwood (L-H) equation of heterogeneous reaction kinetics with a low adsorption constant (K) of 0.07 μM^−1^ (Table 1). Therefore, K × C_0_ in the denominator can be ignored in the realm of low concentrations. The L-H equation thus becomes a simple pseudo-first-order linear kinetic equation (r_0_ = k_L−H_ × K × C_0_ = 0.3 C_0_, μm min^−1^), which indicates the involvement of a solid-phase interface reaction mechanism in the oxidation of As(III) in the Fe(III)-CaSO_3_ system.

### 2.5. Contribution of Free Radicals to As(III) Oxidation

Several studies have shown that the generation of reactive oxygen species is the main reason for the oxidative degradation of pollutants in acidic environments [31,32]. The Fe(III)-CaSO_3_ system involves oxysulfur free radicals, including SO_3_^•−^, SO_5_^•−^, and SO_4_^•−^, formed according to Equations (1)–(8). HO^•^ may be generated from SO_4_^•−^, according to Equations (21) and (23), and may also contribute to the degradation of As(III). Quenching experiments were performed to better understand the reaction mechanisms (Figure 5). Ethanol (EtOH) and tert-butyl alcohol (TBA) were employed as radical scavengers.
SO_4_^•−^ + H_2_O → SO_4_^2−^ + HO^•^ + H^+^(23)

EtOH has similar rate constants for SO_4_^•−^ and HO^•^ (k_EtOH,HO_^•^ = (1.8–2.8) × 10^9^
M^−1^ s^−1^, k_EtOH,SO4_^•−^ = (1.6–7.7) × 10^7^
M^−1^ s^−1^) [33]. However, the rate constant for the reaction between TBA and HO^•^ (k_TBA,HO_^•^ = (3.8–7.6) × 10^8^ L mol^−1^ s^−1^) is 1000-times higher than that for the reaction between TBA and SO_4_^•−^ (k_TBA,SO4_^•−^ = (4.0–9.1) × 10^5^ L mol^−1^ s^−1^) [34]. Hence, SO_4_^•−^ and HO^•^ contributions for As(III) oxidation can be distinguished by adding EtOH and TBA, respectively. After adding EtOH (5 mM), it was found that the oxidation rate of As(III) decreased from 0.4 to 0.07 min^−1^, implying that this main reactive species generated by Fe(III)-CaSO_3_ process were attributable to SO_4_^•−^ and HO^•^, the residual should be caused by SO_5_^•−^, as SO_3_^•−^ possesses so weak oxidative ability that cannot oxidize As(III) and SO_3_^•−^ tends to be rapidly oxidized to SO_5_^•−^ in the oxygen-containing condition [28]. Therefore, we surmised that SO_5_^•−^ caused the 17.5% As(III) oxidation. Moreover, when adding TBA (2 mM) into the solution, the oxidative rate decreased to 0.13 min^−1^, showing that HO^•^ was responsible for about 67.5% of the As(III) oxidation. In conclusion, the reactive species mainly responsible in the Fe(III)-CaSO_3_ system involved SO_4_^•−^, HO^•^ and SO_5_^•−^ generation that accounted for 15%, 67.5% and 17.5% contribution for As(III)oxidation, respectively.

## 3. Materials and Methods

### 3.1. Materials

NaAsO_2_ (99.5%; Gracia Chemical Technology Co., Ltd., Chengdu, China) was dried in a desiccator for 24 h prior to use. Na_2_HAsO_4_·7H_2_O was purchased from Alfa Aesar (A Johnson Matthey Co., Ltd., Shanghai, China). CaSO_3_ was purchased from Shanghai Aladdin Bio-Chem Technology Co., Ltd. (Shanghai, China). Fe_2_(SO4)_3_, NaOH, H_2_SO_4_, KBH_4_ and HCl were purchased from Sinopharm Chemical Reagent Co., Ltd. (Shanghai, China). Ethanol (EtOH) and tert-butyl alcohol (TBA) were used for radical quenching. All chemicals were of analytical reagent grade or higher purity, and were used without further purification.

### 3.2. Reaction Procedure

All experiments were performed in a 250 mL cylindrical reactor cooled by external jacket water circulation at a constant temperature of 25 °C (Figure 6). A 250 mL solution containing As(III) at the desired concentration was placed in the reactor and constantly stirred with a polytetrafluoroethylene-coated magnetic stirrer. Sulfite consumed oxygen quickly in aqueous solution, so hence synthetic air was constantly pumped into the reaction solution. Then, Fe(III) and CaSO_3_ solutions were added, and the pH was quickly re-adjusted to the desired value (± 0.1). Aliquots (4.5 mL) were withdrawn at specific time intervals and immediately mixed with 0.5 mL portions of 6 M HCl. The resultant mixtures were filtered through a 0.22 μm filter and preserved in the dark at low temperature (4 °C) for less than 4 h before analysis. The concentrations of As(III) and As(V) in the filtrates were determined by HPLC-HG-AFS.

### 3.3. Analysis

The sample was acidified and filtered in advance, so that the adsorbed As in the reaction solution has basically transformed it into dissociative state. Arsenic concentration was determined by using liquid-phase hydride-generation–atomic fluorescence spectrometry (LC-HG-AFS). As(III) and As(V) in the reaction solution were separated on an ion chromatography column (PRP-X100, Hamilton, Switzerland) by eluting with a phosphate mobile phase (45 mM, pH 5.6). The concentrations of As(III) and As(V) were determined using 5% HCl–2% KBH_4_ solution in HG-AFS [29].

## 4. Conclusions

The Fe(III)-CaSO_3_ system can generate free radicals and effectively degrade As(III). It has been demonstrated that hydroxyl and oxysulfur radicals are the active species in the mechanism of As(III) oxidation. Under conditions of pH 6.0, the optimal concentrations of Fe(III) and CaSO_3_ in the Fe(III)-CaSO_3_ system are 0.5 and 30 mM, respectively, whereupon the As(III) oxidation efficiency can reach 72.1% after 1 h. Radical scavenging tests have indicated that oxidation of As(III) is caused by HO^•^ (67.5%), SO_5_^•−^ (17.5%) and SO_4_^•−^ (15%). The results presented here imply that in the Fe(III)-CaSO_3_ system, replacing soluble sulfites with slightly soluble sulfites is an effective strategy in oxidizing As(III) to As(V), and in the Fe(III)-CaSO_3_ system, replacing soluble sulfites with slightly soluble sulfites is an effective in degrading arsenic, and the Fe(III)-CaSO_3_ system offers a cost-effective process for arsenic removal from contaminated waters, and that sulfite holds significant promise as a new resource utilization method of desulfurized gypsum.

## Figures and Tables

**Figure 1 molecules-26-01154-f001:**
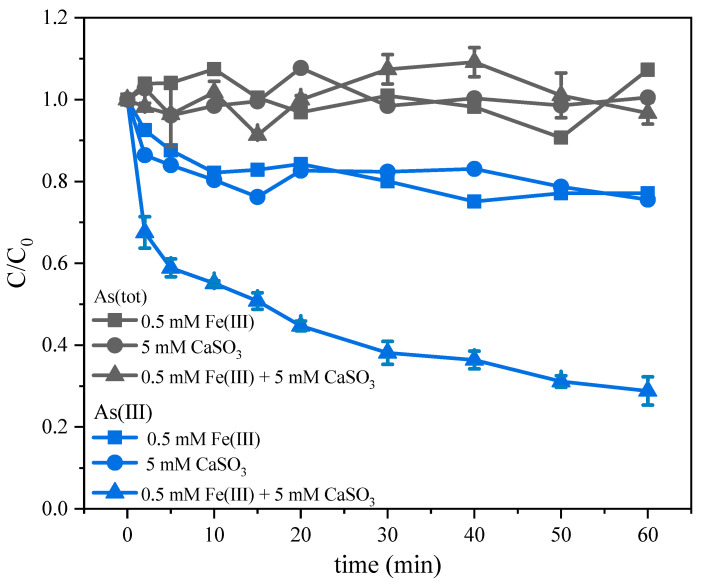
Concentration changes of As(III) and As(tot) in the control experiments of Fe(III)-CaSO_3_ system. Initial conditions: [Fe(III)] = 0.5 mM, [CaSO_3_] = 5 mM, [As(III)] = 5 μM, pH 6.0, air bubbling at 0.5 L min^−1^.

**Figure 2 molecules-26-01154-f002:**
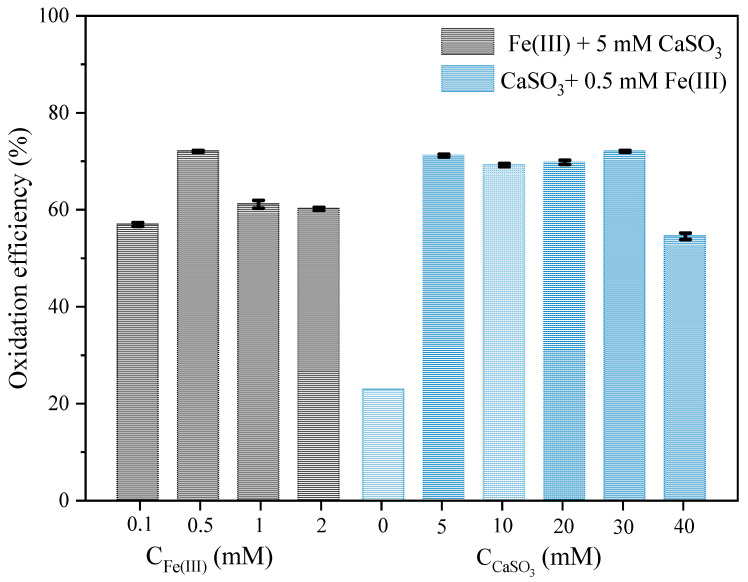
Effect of dosage of Fe(III)/CaSO_3_ on the oxidation efficiency of As(III) in the Fe(III)-CaSO_3_ system after a reaction time of 60 min. Initial conditions: [As(III)] = 5 μM, pH 6.0, air bubbling at 0.5 L min^−1^. Fe(III) + 5 mM CaSO_3_: [Fe(III)] = 0.1–2.0 mM, [CaSO_3_] = 5 mM; CaSO_3_ + 0.5 mM Fe(III): [Fe(III)] = 0.5 mM, [CaSO_3_] = 0–40 mM.

**Figure 3 molecules-26-01154-f003:**
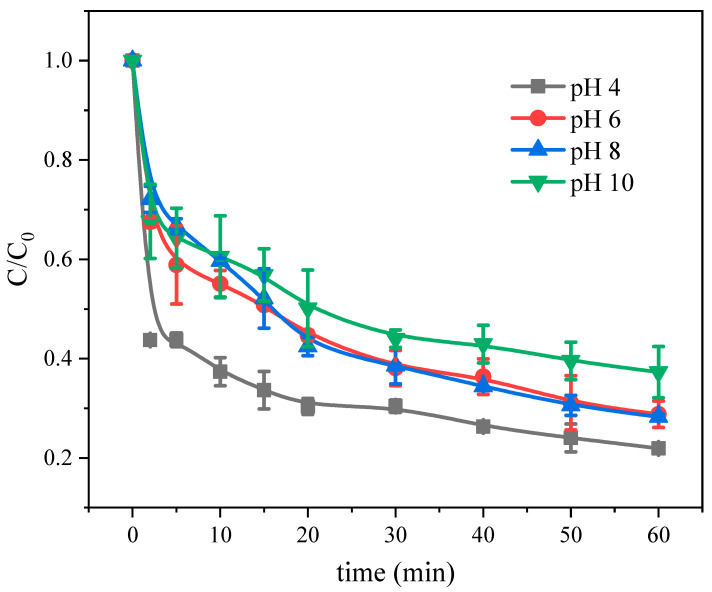
Effect of pH on the Fe(III)-CaSO_3_ system. Initial conditions: [Fe(III)] = 0.5 mM, [CaSO_3_] = 5 mM, [As(III)] = 5 μM, air bubbling at 0.5 L min^−1^.

**Figure 4 molecules-26-01154-f004:**
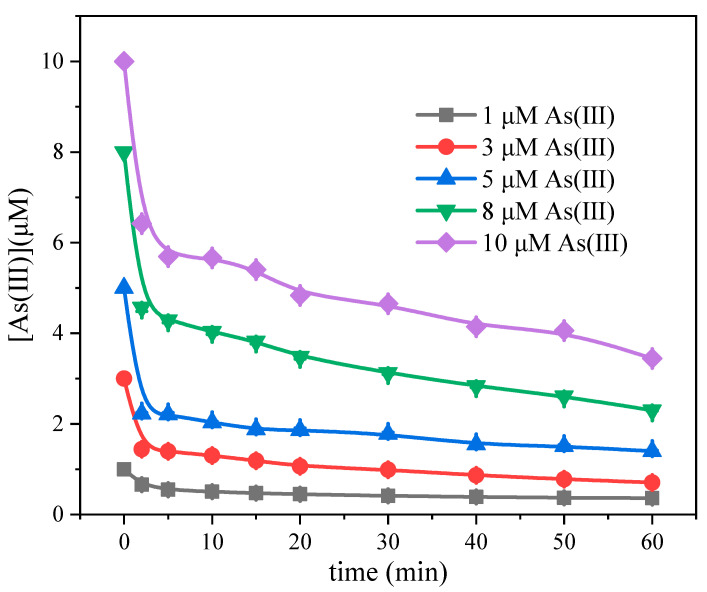
Effect of initial As(III) concentration on the Fe(III)-CaSO_3_ system. Initial conditions: [Fe(III)] = 0.5 mM, [CaSO_3_] = 30 mM, [As(III)] = 1–10 μM, pH 6.0, air bubbling at 0.5 L min^−1^.

**Figure 5 molecules-26-01154-f005:**
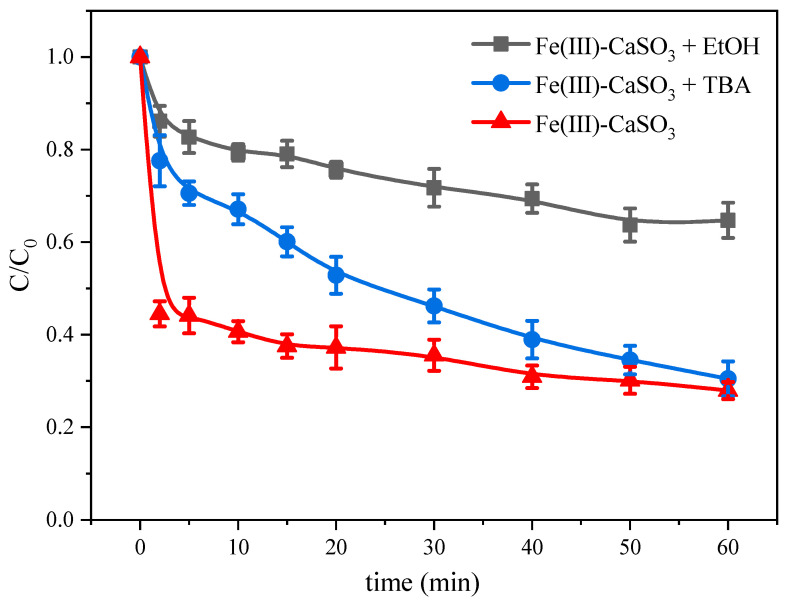
Contribution of free radicals in the Fe(III)-CaSO_3_ system. Initial conditions: [Fe(III)] = 0.5 mM, [CaSO_3_] = 30 mM, [As(III)] = 5 μM, [EtOH] = 5 mM, [TBA] = 2 mM, pH 6.0, air at 0.5 L min^−1^.

**Figure 6 molecules-26-01154-f006:**
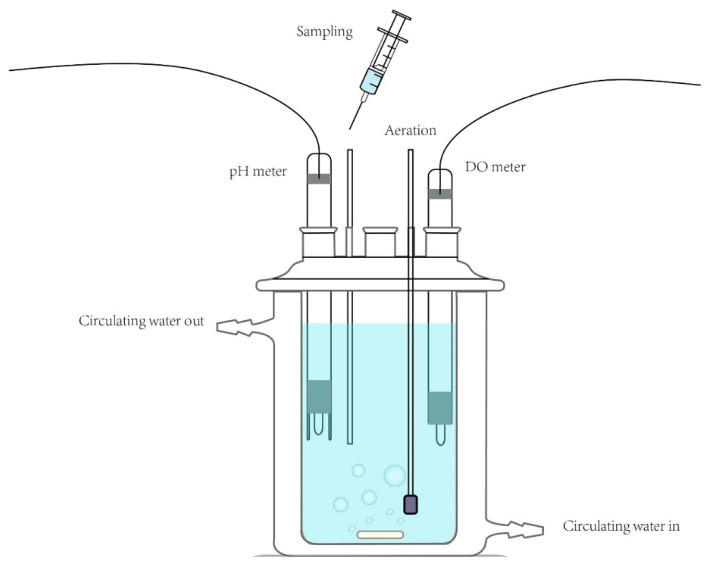
Schematic of experimental setup.

**Table 1 molecules-26-01154-t001:** Oxidation kinetics of As(III) according to the L-H equation in the Fe(III)-CaSO_3_ system.

C_0_	r_0_	Kinetic Equation	k_L−H_	K	R^2^
1	0.1694	$r_{0}=\frac{0.3282c}{\left(1+0.07382c\right)}$	4.446	0.074	0.972
3	0.6731				
5	1.3876				
8	1.7167				
10	1.7896				

C_0_ is initial concentration (μM); r_0_ initial rate (μM min^−1^), k_L−H_ the rate constant (μM min^−1^), and K the adsorption constan (μM min^−1^).

## Data Availability

Data is contained within the article.

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
