# Peer review of "Calcium Sulfite Solids Activated by Iron for Enhancing As(III) Oxidation in Water"

_molecules, 2021, doi:10.3390/molecules26041154_

Round 1

Reviewer 1 Report

Dear Authors,

The article entitled “Calcium sulfite solids activated by iron for enhancing As(III) oxidation in water”, in my opinion, is suitable for publication in the journal “Molecules” and can be accepted. Only limited number of minor comments have been formulated in the review.

Details of the review can be found below.

Title
Accurate.

Abstract
Accurate.

Keywords
Accurate.

Introduction
Accurate.
Introduction section gives sufficient description of the research issue on the application background.
Minor remarks:
Row 28:          Typing error “gyps_um” must be corrected as “gypsum”.

Materials and methods
Accurate.
There are no substantial objections to description of methodology besides 1 minor comment:
Rows 71-73:   Sampling and preserving procedures must be more clearly described.

Results and discussion
Acceptable.
Explanations for process mechanisms have occasionally a speculative character, however they are consistent and well support conclusions.
Minor remarks:
Rows 130-131:           The sentence (rows: 128-131) is rather too complex and the last part seems to be unclear. The relationship between S(IV) and As(III) concentrations with the factor of “8000 times” must be clarified.
Row 140:        Should be – “efficiency varied from …”
Table 1:           The equation must be closed with a bracket “)”.
Row 189:        There is lack of space within the phrase “condition[26]”.

Conclusions
Accurate.

References
Accurate.

Figures and Tables
Accurate.

Reviewer 2 Report

The paper is well written and clear. A schematic of overall experiment would be helpful for the readers.

Strengths:

1. Given, arsenic poisoning is a global health problem, finding new avenues to remove arsenic is always beneficial. This research presents approach of generating free radicals using CaSO3 to degrade arsenic.

2. Material and method section is clear and explicit.

3. Results and discussion are well explained.

4. Toxic mechanisms of arsenic are complex and not fully understood but this paper has provided proposed chemical mechanism where required.

5. The study is well planned that investigates the effect of pH, initial concentration on the Fe(III)-CaSO3 system.

6. Quenching experiments performed adds in important details and finding to the research.

Weakness:

1. To study effect of the initial concentration of arsenic on its oxidation, the concentration chosen was 1μM- 10μM. How and why is this range chosen?

2. For, the free radical being proposed in the chemical reactions, did the authors measure any of those concentration? For e.g interconversion of SO4.- to HO. . How are these concentrations being measured?

3. Schematic of experimental setup could be helpful.

Reviewer 3 Report

The manuscript is interesting and suitable for Molecules.

I suggest only minor revisions, regarding:

1)English grammar double-check of the whole manuscript;

2)better explanation about the math models used for data interpretation;

3)Did the authors perform some cyclyc uses of the material for subsequent As(III) oxidation? The re-usability has been investigated?

4)the authors should provide more comparisons with other materials present in literarature for As(III) oxidation;

5)the authors reported that As(III) has been removed, but actually it is oxidized and then only As(V) can be removed by sorption process. What about the desorption of As(V)?

6)I suggest to the authors to report some more references about the use of iron-based materials for environmental processes in the introduction part to enlarge the state of the art: refer for instance to Chemical Engineering Transactions, 2016, 47, pp. 55-60 and Bulletin of Environmental Contamination and Toxicology, 2018, 101(6), pp. 692-697.

7)conclusion section should better underline the advancement of knowledge reached by the findings of this study.
